# Dairy: Friend or Foe? Bovine Milk-Derived Extracellular Vesicles and Autoimmune Diseases

**DOI:** 10.3390/ijms252111499

**Published:** 2024-10-26

**Authors:** Hairui Ou, Tamas Imre Csuth, Tamas Czompoly, Krisztian Kvell

**Affiliations:** 1Department of Pharmaceutical Biotechnology, Faculty of Pharmacy, University of Pecs, 7624 Pecs, Hungary; hairui.ou@pte.hu (H.O.); tcsuth@foss.dk (T.I.C.); kvell.krisztian@pte.hu (K.K.); 2Soft Flow Ltd., 7634 Pecs, Hungary

**Keywords:** autoimmune diseases, extracellular vesicles, cytokines, bovine milk, miRNA

## Abstract

Due to the availability, scalability, and low immunogenicity, bovine milk-derived extracellular vesicles (MEVs) are increasingly considered to be a promising carrier of nanomedicines for future therapy. However, considering that extracellular vesicles (EVs) are of biological origin, different sources of EVs, including the host origin and the specific cells that produce the EVs, may have different effects on the structure and function of EVs. Additionally, MEVs play an important role in immune regulation, due to their evolutionary conserved cargo, such as cytokines and miRNAs. Their potential effects on different organs, as well as their accumulation in the human body, should not be overlooked. In this review, we have summarized current impacts and research progress brought about by utilizing MEVs as nano-drug carriers. Nevertheless, we also aim to explore the possible connections between the molecules involved in cellular immunity, cytokines and miRNAs of MEVs produced under different health conditions, and autoimmune diseases.

## 1. Introduction

Extracellular vesicles (EVs) are phospholipid bilayer-coated nanoparticles secreted from almost all types of cells, carrying various molecules [1], such as deoxyribonucleic acid (DNA), ribonucleic acid (RNA), proteins, lipids, metabolites and glycans, reflecting the identity and molecular state of their original cell [2]. EVs have biological effects on adjacent and remote target cells by mediating intercellular communication. Additionally, they are emerging as biomarkers and therapeutic agents [1].

The traditional classification of EVs was based on biogenesis (e.g., exosome, microvesicle, apoptotic bodies) [3]. However, there is a trend towards a more detailed classification by function (e.g., autophagic EV, oncosome, matrix vesicle, stress EV, and migrasome) [4,5,6,7] and size (e.g., small oncosome and large oncosome) [1] (Table 1). 

Exosomes share an endocytic origin. They are generated by inward endosome budding to form multivesicular bodies and fusion with the plasma membrane [8]. Microvesicles are a heterogeneous group of membrane-enclosed vesicles formed through direct outward budding and scission from the plasma membrane into the extracellular space [9,10]. Apoptotic bodies arise through the process of apoptosis [11]. These three classical EV types have been thoroughly studied. However, further EV types have been proposed, and the boundaries between them have become blurred.

The EV surface is surrounded by a coronal layer composed of lipids, proteins and glycans [12], which are canonical to EVs, and have great biological significance in regulating EV targeting, homing, and uptake by recipient cells [13,14,15,16], thus playing an important role in their biological distribution [17,18]. Accordingly, removal of this corona has been shown to reduce their uptake and function in different tissues, indicating their importance in facilitating adsorption and subsequent events [19].

Bovine mastitis, primarily caused by bacterial invasion, is an inflammation of the udder tissue in the mammary glands. It manifests as either easily identifiable clinical mastitis or less obvious subclinical mastitis [20]. It was reported that considerable changes occur in the milk proteome during mastitis, following *Staphylococcus aureus* infection [21].

Recent studies indicate that proteins associated with neutrophil extracellular trap (NET) formation are found at higher concentrations in milk from cows infected with mastitis [21]. NETs are fibrous chromatin-based structures composed of granules and nuclear neutrophil constituents [22], where neutrophil phagocytosis and oxidative burst are impaired by milk fat and proteins [23]. 

The modification of EV components, including surface corona and cargo, may be attributed to the formation of NETs. They are also capable of upregulating the expression of several pro-inflammatory factors, thus promoting the activation of the nuclear factor kappa B (NF-κB) pathway [24,25].

It is worth noting that industrial heat treatment of bovine milk has a great impact on the integrity and composition of EVs. Ultra-high temperature (UHT) treatment of milk results in a substantial loss of Evs, while pasteurization hardly affects EV numbers, although it has a negative effect on their integrity and cargo [26]. 

## 2. Milk-Derived EV Isolation

Milk is a rich and complex source of nutritional and immunological components, including milk fat globules, casein micelles, cellular debris, soluble molecules, and Evs [27]. To enrich Evs, other milk components with EV-like characteristics (e.g., density and size) should be removed in advance. For example, the size of casein micelles overlaps with that of Evs, and casein micelles must be depleted before EV isolation [28]. Casein micelles can be precipitated by centrifugation after acidification of milk to pH 4.6 [29,30], aggregated by enzymatic treatment [31], or dissociated by sequestration of calcium with ethylenediaminetetraacetic acid (EDTA), sodium citrate, etc. [32]. Currently, there is no preferred method, but EDTA or sodium citrate are most commonly used. After pre-processing, the clarified milk supernatant can be stored until EV isolation by differential ultracentrifugation (dUC) [33], size-exclusion chromatography [34], or other methods. These can also be combined to improve purity further. The basic process of removing casein using sodium citrate and using dUC is shown in Figure 1A, and the bovine milk-derived extracellular vesicles (MEVs) obtained by this method are shown in Figure 1B.

Further methods are listed below. Density gradient centrifugation [35] applies a density gradient (e.g., sucrose or iodixanol) to separate EVs from other particles. Polymer-based precipitation [36] (e.g., ExoQuick), including the use of polyethylene glycol (PEG), may be added to milk, causing EVs to precipitate, followed by their collection applying low-speed centrifugation. Ultrafiltration [37] involves the use of molecular-weight cut-off membranes to separate EVs based on their size and molecular weight. Immunoaffinity capture [38] is generally integrated with analytic tests such as enzyme-linked immunosorbent assay (ELISA) or magnetic isolation [39]. The advantages and disadvantages of the above methods are shown in Table 2.

## 3. Autoimmune Diseases and Milk Consumption

Autoimmune diseases are characterized by the immune system mistakenly attacking self-molecules as a result of a breakdown of immunologic tolerance to self-antigens [40]. This occurs as the immune system loses its ability to distinguish between self and non-self. Many autoimmune disorders are triggered by genetic predisposition, microbial and environmental factors [41].

Two major phenotypes, ulcerative colitis (UC) and Crohn’s disease (CD), constitute inflammatory bowel disease characterized by chronic, fluctuating inflammation of the intestine [42]. More specifically, UC is an inflammatory disease primarily targeting the mucosal lining of the rectum and large intestine. In contrast, CD is an inflammatory disorder affecting the entire digestive tract, from mouth to anus, resulting in lesions between areas of healthy tissue [43,44].

Rheumatoid arthritis (RA) is the most common inflammatory arthritis, which primarily targets the small joints and tissue lining, leading to pain, deformities, and impaired function [45]. The similarity between certain self and foreign antigens (molecular mimicry) has been proposed to explain the autoimmune response in this disease [46].

According to the multiple hit model of RA development, environmental and genetic factors are both needed for the development and progression of the disease, to which process the consumption of bovine milk may also contribute [47]. Dietary allergens in childhood, especially early exposure to milk proteins, may have a possible impact on the onset of future diseases [48]. The serum antibodies of RA patients recognize bovine albumin, and bovine serum albumin (BSA) shows homology with human antigens [49]. In an RA model, a close correlation was found between disease-related and milk-specific antigens. Both immunoglobulin (Ig) G and IgE antibody levels were found to be elevated in serum [50]. Interestingly, an inverse association was found between the incidence of RA and vitamin D [51]. 

Intestinal dysbiosis was also found to be important in the development of RA [52]. The intestinal epithelial cells form a mucosal barrier, which functions as defense against bacterial invasion and infection [53]. MEVs were found to participate in the regulation of the proliferation of intestinal epithelial cells and the development of the digestive tract [54]. Experimental results show that MEVs upregulate the expression of phosphorylated extracellular signal-regulated kinases (ERK) and p-38 in an intestinal cell line. These proteins are part of the mitogen-activated protein kinase (MAPK) pathway, which is important in cell growth, proliferation, differentiation and apoptosis [55]. 

Type 1 diabetes (T1D) is an autoimmune disease likely resulting from a complicated interplay of genetic predisposition, epigenetic, and environmental factors. Ecological studies have identified temporal and geographical correlations between T1D incidence and bovine milk consumption [56,57].

Multiple sclerosis (MS) is considered to be a chronic autoimmune disease arising from a combination of genetic factors and early-life environmental triggers, which are thought to disrupt immunological self-tolerance to myelin in the brain and the spinal cord [58]. Epidemiological studies have linked the prevalence of MS with the consumption of milk and dairy products [59,60].

Molecular mimicry also plays a part in the development of MS, as butyrophilin, a common milk protein, can induce T-cell response against self-antigen myelin oligodendrocyte glycoprotein [61]. Following milk consumption, antigen-specific immune responses are induced by the butyrophilin-derived peptides that cross the gut mucosa. These responses in the gut-associated lymphoid tissue and peripheral immune organs normally induce tolerance, but can also induce an inflammatory T-cell response [62]. 

### 3.1. Autoimmune Diseases and Cytokines

Cytokines are a diverse group of small, secreted proteins, crucial in mediating intercellular communication, modulating immune responses, and maintaining health status or disease. Cytokines can serve as both pro- and anti-inflammatory agents. In addition, they can also act as immune stimulants for the development and maturation of the immune system [63,64,65].

It was shown that cytokines may be released in both soluble and EV-associated forms. Among cytokines associated with EVs, interleukin (IL)-2, IL-4, IL-10, IL-12, IL-15, IL-16, IL-18, IL-21, IL-22, IL-33, Eotaxin, interferon gamma-induced protein (IP)-10, inducible T-cell alpha chemoattractant (ITAC), macrophage colony-stimulating factor (M-CSF), monokine induced gamma interferon (MIG), macrophage inflammatory protein (MIP)-3α, transforming growth factor (TGF)-β, and tumor necrosis factor (TNF)-α, are preferentially encapsulated in EVs across all systems [66]. There are multiple biological reasons to load EVs with cytokines. It was proposed that EV-entrapment is a mechanism to dispose of pathogenic or oncogenic products [67,68]. It can also facilitate cytokine delivery to distant cells. Most probably, EVs protect cytokines from environmental degradation as well, such as trypsin digestion [66].

Healthy bovine milk predominantly contains several immune regulatory cytokines, including IL-4, IL-10, interferon (IFN)-γ, and TGF-β [69,70]. However, in the case of mastitis, the presence of cytokines such as IL-4, IL-6, IL-12, IL-13, IL-17A, TNF-α, and IFN-γ is more pronounced [71]. Of note, cytokines are evolutionarily conserved and may be effective across species [72]. Moreover, active TGF-β was found in these vesicles, which, under inflammatory conditions, can induce T helper (Th)17 cells. Uncontrolled Th17 activation has been linked with the development of many autoimmune diseases [73].

It is suggested that the Th2 immune response, with inflammation and epithelial barrier dysfunction, characterizes UC [74], which has a possible connection with IL-4 [75] and IL-10 [76]. CD, in contrast, is driven by a Th1/Th17 response in which IL-12 plays a key role [77]. Increased level of TNF protein in the intestine is observed in CD [77], and neutralization of active human TNF protein reduces recruitment of inflammatory cells and granuloma formation in humans [78]. In accordance with this, anti-human TNF protein monoclonal antibody is often used as treatment [79]. Moreover, deregulated TGF-β signaling is observed in the intestines of IBD animals and patients [79,80]. However, IL-10 is regarded as a major regulatory cytokine associated with many autoimmune diseases, including chronic inflammatory bowel disease (IBD), due to its capability to inhibit both antigen presentation and consecutive pro-inflammatory cytokine release [81]. 

In patients with active RA, IL4, as a protective angiostatic cytokine, was significantly elevated when compared to healthy individuals [82]. However, TNF-α is widely recognized as the primary inflammatory cytokine involved in the development and progression of RA, and blockade of TNF-α can reduce the severity of disease in mammals, including humans [78,83]. Similarly, IL-12 can induce the production of pro-inflammatory cytokines, which may be linked to increased RA severity [84]. To compensate for inflammation, elevated levels of IL-10, both in duodenal messenger RNA (mRNA) and serum protein, are also associated with RA in humans [85,86]. Additionally, TGF-β protein has been shown to exacerbate RA in animal models [87,88]. 

Up-regulation of TGF-β in the renal cortex has been associated with T1D in *OVE26* diabetic mice [89]. In *BDC2.5* T cell receptor transgenic mice, TGF-β is necessary for the development of T1D [90], and up-regulation of TGF-β mRNA in renal glomeruli is associated with the disease in mice [91]. TNF-α is also involved in the pathogenesis of T1D [92], with specific mutations in the human *TNF gene* (e.g., *rs1799964*) being linked to T1D [93]. Additionally, IL-12 proteins have been shown to exacerbate insulin-dependent diabetes mellitus in non-obese diabetic (NOD) mice over-expressing human leukocyte antigen-DR isotype (HLA-DR) alpha protein [94]. 

It has been reported that the serum levels of IL-4 in patients with MS are nearly three times higher than those in healthy controls, with no significant difference between males and females [95], indicating that human IL4 protein in serum is associated with the active form of MS [96]. Conversely, down-regulation of TGF-β in serum is associated with the fluctuating form of MS [97]. Compelling evidence from studies in both human and experimental models of MS has demonstrated that TNF-α is involved in several key pathological hallmarks of the disease, including demyelination, synaptopathy, and neuronal inflammation [98]. Meanwhile, increased TNF-α mRNA levels in mononuclear cells from cerebrospinal fluid are linked to the disease [99]. Furthermore, the up-regulation of IL-10 mRNA in peripheral blood mononuclear cells is also connected to MS [99]. The IL-10/IL-10R axis has been identified as a crucial mechanism for constraining inflammation during MS [100].

The summarized relationship between cytokines of MEVs and autoimmune diseases is shown in Figure 2.

### 3.2. Autoimmune Diseases and miRNAs

Micro ribonucleic acids (miRNAs) are small (19–22 nucleotides long) RNAs that regulate gene expression at the post-translational level [101]. miRNAs cause degradation and/or translation repression by binding to complementary sequences of target mRNAs [102]. The expression of at least 60% of human genes is regulated post-transcriptionally by miRNAs, contributing to many cellular processes like cell proliferation, differentiation and cell death [103]. Microarray analysis revealed the miRNA profile of MEVs, and some of the most highly expressed miRNAs have a clear connection to autoimmune diseases [104]. The list of abundant miRNAs in MEVs is shown in Table 3.

miRNAs could contribute to the development of autoimmune inflammatory responses in many ways, including the activation of antigen-presenting cells, antigen recognition by specific lymphocyte receptors, differentiation of different T-cell subsets, altering the functions of regulatory T Cells (T-reg) cells, cytokine production, and the recruitment of inflammatory cells by cytokines and chemokines [105]. Microarray analysis revealed the miRNA profile of exosomes isolated from healthy bovine milk [104]. miR148a, miR-375, miR-141, miR-200a and miR-223 are among the most abundant miRNAs in milk-derived exosomes, and could be related to autoimmune diseases. The expression levels of miRNAs remain high through the lactation period, suggesting their functionality and importance [106]. Interestingly, miR148a is among those abundant miRNAs whose sequence and presence in several mammalian milks (human, porcine, bovine, and panda) is conserved [107].

**Table 3 ijms-25-11499-t003:** List of abundant miRNAs in MEVs [104,106]. The miRNAs potentially connected with autoimmune diseases are in bold.

Highly Expressed miRNAs in MEVs				
miR-2478	miR-2328	miR-29c	miR-26b	let-7d
miR-1777b	let-7c	miR-30d	miR-2892	**miR-200a**
miR-1777a	**miR-148a**	miR-92	miR-24-3p	miR-151
let-7b	miR-320	miR-2304	miR-1249	miR-30f
miR-142a	miR-2888	**miR-375**	miR-423-5p	miR-2291
miR-2412	let-7f	miR-2391	miR-664	miR-2284l
miR-2305	miR-200b	let-7g	miR-2887	miR-21e5p
let-7a	miR-1584	miR-30a-5p	miR-2284d	miR-30c
miR-200c	miR-26a	miR-1343	**miR-223**	miR-22e3p
**miR-141**	miR-20a	miR-125b	miR-2374	miR-30e-5p
miR-2881	miR-103	miR-30b-5p	miR-29a	miR-423e5p

Mutai et al. found that the plasma levels of some immune-related miRNAs increased after milk consumption, suggesting that MEVs are bioavailable in humans after the consumption of milk [108]. 

Host immune response to infection changes the miRNA content of MEVs as well. Specifically, miR-142a and miR-223 were found as potential biomarkers for early detection of mastitis [109]. miR-142a is involved in neuronal inflammation in MS, suppressing protective genes such as *TGFβR1* and *SOCS1*, resulting in a shift of T-cell differentiation towards IFN1-producing Th1 cells [110]. Additionally, miR-223 could be a biological marker in RA, as it has a regulatory role by increasing the sensitivity of macrophages to pro-inflammatory cytokines [111].

miR-141 and miR-200a were found to be up-regulated in MS patients, and as a result, the percentage of Th17 cells increased in parallel with a decrease in T-reg cell levels. These miRNAs are involved in T-cell differentiation, most probably through Janus kinase/signal transducers and activators of transcription (JAK/STAT), TGF-β and mammalian (or mechanistic) target of rapamycin (mTOR) pathways [112]. The pathogenic subset of Th17 is a potent inducer of autoimmune inflammation [113]. In silico analysis identified *EGR2* as the target of miR-141 and miR-200a. Their inhibition leads to down-regulation of *SOCS3*, both being important in the limitation of Th17 differentiation [114]. 

miR-148a is an important regulator of B-cell tolerance, and its deregulation leads to the development of autoimmune diseases. With the suppression of *Gadd45a, PTEN* and *Bcl2l11*, miR-148a protects immature B-cells from apoptosis activated by B-cell receptor engagement [115]. miR-148a, after transcriptional activation by PU.1, also promotes monocyte-derived dendritic cell differentiation through the direct inhibition of MAF BZIP Transcription Factor B (MAFB), and this process results in immune imbalance and inflammatory response in psoriasis [116]. 

miR-375 level was found to be elevated in the serum of patients with autoimmune thyroid autoimmune diseases. miR-375 regulates thymic stromal lymphoprotein (TSLP), which promotes the differentiation of Th17 cells in Graves’ disease [117]. Circulating miR-375 levels showed a rapid increase in mice after the induction of B-cell death with streptozotocin toxin, suggesting that miR-375 might be a potential biomarker for T1D [118]. In IBD, miR-375 is one of the six miRNAs that might be used to discriminate UC and CD [119]. The up-regulation of miR200a and miR-141 was also reported in IBD [120]. 

In summary, MEVs naturally harbor miRNAs that play a role in the development of autoimmune diseases. However, a direct causal link between milk consumption and the onset of these illnesses has not been definitively established. While miRNAs from MEVs appear to be biologically accessible to human cells, the critical concentration required for triggering autoimmune processes through milk consumption remains uncertain. Given the multifactorial nature of autoimmune disease development, it is plausible that miRNAs may indeed contribute to these complex processes.

## 4. Extracellular Vesicles as Potential Therapeutic Agents

Since MEVs withstand the hostile conditions of the gastrointestinal system, EVs can protect their internal miRNA cargo against degradation by RNases and low pH [121]. 

In intestinal cell models, MEVs showed beneficial uptake properties compared to liposomes, although the uptake mechanism of EVs into intestinal cells is not completely clear yet [122]. The uptake of vesicles by the intestinal cell line Caco-2 was examined using fluorescence-labeled MEVs. It was found that the cells internalize vesicles in a time- and dose-dependent manner, and more importantly, the EVs had no deleterious effect on cell viability [123]. MEVs were found to improve gut barrier integrity in chemical-induced colitis as well, by increasing the proportions of T-reg cells and macrophages in the intestine, reinforcing epithelial tight junctions, and increasing mucin secretion [124]. Oral administration of MEVs in the colitis mouse model decreased intestinal inflammation by the inhibition of Toll-like receptor 4 (TLR4)- NF-κB and NOD-, LRR- and pyrin domain-containing protein 3 (NLRP3) pathways and restoration of the balance between T-reg and Th17 cells [125]. There are implications that dietary MEVs can accumulate in resident macrophages of the liver and spleen [126]. Moreover, MEVs were found to cross the blood–brain barrier, demonstrating that these vesicles have the potential to regulate gene expression in various tissues and possess systemic effects [127].

Similarly, a mouse macrophage cell line Raw364.7 takes up MEVs without any sign of cytotoxicity, supported by in vivo experiments also lacking any systematic toxicity or elevation of inflammatory cytokines [128]. Curcumin encapsulated in EL-4 cell-derived EVs had five-fold higher solubility compared to free curcumin, and its stability also increased significantly [129]. 

EV encapsulation of nucleic acid was also successful; EVs protected exogenous siRNA from degradation in harsh conditions, and made the cargo available for cellular uptake in Caco-2 cells [130]. Cargo loading into EVs can be either a passive or an active process. Passive loading means the incubation of EVs or the producer cells with the bioactive compound, which accumulates the compound [131]. Active cargo loading involves methods like sonication, electroporation, extrusion, incubation with membrane permeabilizers, and the use of freeze–thaw cycles, which allow the drug to diffuse through the compromised membrane integrity of EVs [132]. Raw bovine milk could serve as a biocompatible source for harvesting big quantities of EVs in a cost-effective way [133]. Up to date, there are no existing solutions that use EVs as a carrier in the treatment of autoimmune diseases, but with a better understanding of those illnesses, the therapeutic potential of EVs might be used successfully. 

The positive effect of MEVs on the intestinal tract was already mentioned, and there is evidence that they have beneficial properties against other autoimmune diseases as well. Arntz et al. found that orally administered MEVs showed therapeutic effects on RA murine models. EV treatment had an inhibitory effect on innate immunity, as the expression of chemokines monocyte chemoattractant protein-1 (MCP-1), chemokine KC and IL-6 was reduced. The reason behind this might be that MEVs contain bioactive TGF-β and other immune regulatory miRNAs [134]. 

There is proof that MEVs can ameliorate UC. Two subsets of EVs were found to prevent colonic tissue damage and preserve colon barrier integrity, and the vesicles restored high levels of anti-inflammatory protein A20, an inhibitor of NFκB. Moreover, EVs down-regulated the expression of miR-125b, an inhibitor of A20, and restored levels of Zonula Occludens-1 (ZO-1), which is suppressed in digestive inflammation [135]. In the dextran sulfate sodium-induced colitis murine model, MEVs were involved in the regulation of a wide range of proteins, including the significant suppression of pro-inflammatory cytokines (IL-1B, IL-6, IL-17A and IL-33), chemokine ligands (CXCL1-3, CXCL5, CCL3, CCL11) and chemokine receptors (CXCR2 and CCR3), and the promotion of anti-inflammatory gene expression [136]. 

In the colitis murine model, MEVs showed therapeutic and anti-inflammatory effects, which involved highly expressed miRNAs such as miRNA-375, miRNA-let7a, miRNA148, and miRNA-320 and regulatory proteins like TGF-β. One of the targets of miRNAs is DNA methyltransferases (DNMTs), which are regulators of genes controlling epigenetic DNA methylation and involved in the aberrant methylation pattern observed in IBD patients. Following treatment with MEVs, DNMT1 and DNMT3 were down-regulated in the colonic tissue, and, in parallel, TGF-β protein levels were up-regulated [137]. Interestingly, in order to reach the therapeutic dose for the positive effects of MEVs on colitis, an adult might have to drink 1 liter of fresh milk per day, which underlines the importance of the need for proper formulation before possible therapeutic applications [126]. 

The transdermal delivery of MEVs rich in butyrophilin, a protein that shows high homology with myelin oligodendrocyte glycoprotein, might be used in the immunotherapy of MS [138].

In conclusion, although milk may induce autoimmune processes by falsely recognized proteins, MEVs are likely advantageous, through the maintenance of gut barrier functions.

## 5. Challenges and Future Research

Extensive research has explored the role of MEVs in physiological functions and their expanded utility in applications such as drug delivery and therapeutics, yielding promising results. However, many mysteries remain to be solved.

First, it is indispensable to consider the fact that MEVs reflect the condition of the secretory cells from which they originate. Therefore, the composition of MEVs can be influenced by the physiological state of the host where the milk originates [68], making the quality control of MEVs a critical issue that needs to be carefully addressed.

Even though there are now a variety of different isolation methods to extract EVs from milk, the purity level of MEV-isolation solutions is usually not enough to attribute the observed effects to specific vesicles or include other unknown components. Meanwhile, MEVs extracted by different methods may also contain different ratios of various subgroups with diverse biological properties and phenotypes, which may affect subsequent therapeutic applications.

There is a potential risk that MEVs may trigger an immune response in humans, especially if sub-clinical underlying immune conditions are present, given the evolutionarily conserved nature of MEVs and their cargo. An in vivo study on the anticancer properties of MEVs reported that MEVs have the risk of increasing metastasis, depending on the time of treatment [139]. Further studies are required to define the precise effects of these processes on MEVs and their cargo in terms of altered bioavailability and bioactivity.

While, to date, various observations indicate EVs in various biological fluids as mediators of cellular crosstalk, more efforts are now underway to better understand the exact role of their cargo in immune regulation and modulation [128,140,141].

Particularly, it is worth noting that many bovine-derived miRNAs have not yet been annotated, and their potential effect on the human body remains unknown [110]. An enhanced risk of developing allergies, neurodegenerative diseases, obesity, diabetes, and even cancer in later stages of life [142,143] have been primarily attributed to miRNA and persistently elevated mammalian target of rapamycin complex 1 (mTORC1) signal, however, these results come from continued milk consumption during adolescence [144,145]. 

In summary, it becomes imperative to consider the potentially harmful consequences of MEV cargo in eliciting diseases such as autoimmune diseases or promoting disease progression in other cases. The use of MEVs requires comprehensive consideration of the physiological conditions of the donor and the recipient, whose effects may also be context-dependent. Thus, before exploiting MEVs for therapy, it is vital to address safety concerns.

## Figures and Tables

**Figure 1 ijms-25-11499-f001:**
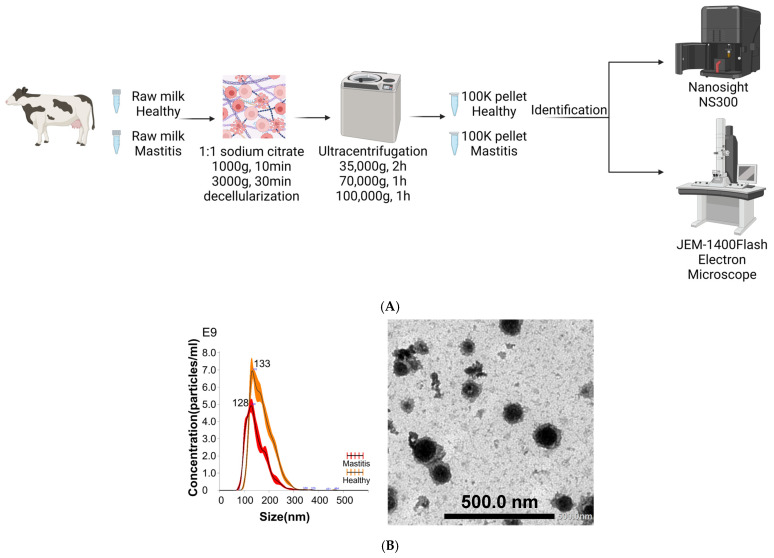
(**A**) shows the isolation steps of bovine milk-derived extracellular vesicles (MEVs) from skimmed bovine milk, followed by identification through morphological characterization with nanoparticle tracking analysis (NTA) and transmission electron microscopy (TEM) (Created with BioRender.com, Toronto, ON, Canada). (**B**) depicts the size distribution of extracellular vesicles from both healthy and mastitis groups, as measured by NTA NS300 (Malvern Panalytical, Malvern, United Kingdom). This analysis revealed that the mode sizes of MEVs in the healthy and mastitis groups are 133 nm and 128 nm, respectively. In addition, the concentrations of MEVs were found to be around 6.38 × 10^11^ ± 9.56 × 10^9^ particles/mL in the healthy group and 4.07 × 10^11^ ± 1.32 × 10^10^ particles/mL in the mastitis group. Additionally, it presents a transmission electron microscope (TEM) image capturing extracellular vesicles for visual reference.

**Figure 2 ijms-25-11499-f002:**
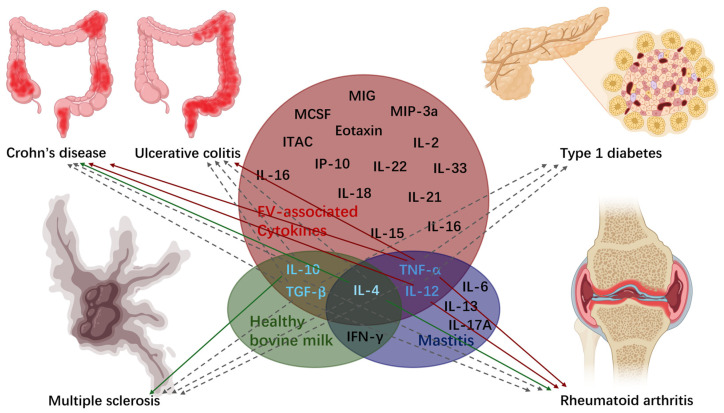
This figure summarizes the relationship between autoimmune diseases and EV-associated cytokines in bovine milk across various health conditions. The diagram uses grey dashed lines to represent association, green lines to indicate inhibition of autoimmune diseases, and red lines to denote the promotion of such diseases in humans (created with BioRender.com, Toronto, ON, Canada).

**Table 1 ijms-25-11499-t001:** Summary of EVs by category, size, markers, and biogenesis.

Category	Size	Markers	Biogenesis
Exosome	30–150 nm	CD9, CD63, CD81	Multivesicular endosome
Microvesicle	150–1000 nm	Integrins, P-selectin	Plasma membrane shedding
Apoptotic bodies	200–5000 nm	Annexin V, Histones	Apoptosis
Autophagic EV	40–500 nm	LC3B, p62/*SQSTM1*	Autophagosome–endosome fusion (Amphisome)
Oncosome	100–400 nm	Phosphatidylserine	Derived from cancer cells
Large oncosome	1–10 μm	Cytokeratin 18	Cellular transformation, abnormal assembly of molecular cargo
Matrix vesicles	30–300 nm	Fibronectin, Alkaline phosphatase	Matrix binding and release
Stress EV	40–1000 nm	Heat shock proteins 70, Heat shock proteins 90	Response to cellular stress
Migrasome	500–3000 nm	Tetraspanin 4/7, Integrins α1, α3, α5, and β1	Cell migration

**Table 2 ijms-25-11499-t002:** Advantages and disadvantages of different EV isolation methods.

Methods	Advantages	Disadvantages
Differential ultracentrifugation	Cost-effective High yield	Damage to EVs Low purity
Size-exclusion chromatography	Enhanced purity Uniform-sized EVs	Low yield Not suitable for high volumes
Density gradient centrifugation	Relatively pure High separation efficiency	Low productivity Time-consuming
Polymer-based precipitation	Simple and fast	Co-precipitation of impurities Expensive for large volumes
Ultrafiltration	Simple	Varying efficiency of filtering due to clogging of membranes Loss of EVs due to dead volume
Immunoaffinity capture	High specificity for EVs	Not suitable for large volumes Expensive, due to the cost of antibodies

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
