# Peer review of "Dairy: Friend or Foe? Bovine Milk-Derived Extracellular Vesicles and Autoimmune Diseases"

_ijms, 2024, doi:10.3390/ijms252111499_

Round 1
Reviewer 1 Report
Comments and Suggestions for Authors
The manuscript provided a brief introduction of extracellular vesicles and a good summary of milk derived EV in terms of its role in autoimmune diseases. The manuscript also reviewed the potential and challenges of milk derived EVs as potential therapeutic agents. The writing is clear and pleasing to read.
I was wondering if authors could include any quantitative data in pertaining studies, e.g. EV concentrations in milk, how much cytokine and miRNA in milk-derived EVs, and provide slightly more in depth details in terms of the relationship of milk-derived EVs/milk consumptions and diseases, e.g. Line 140-141.
Author Response
Comments:
The manuscript provided a brief introduction of extracellular vesicles and a good summary of milk derived EV in terms of its role in autoimmune diseases. The manuscript also reviewed the potential and challenges of milk derived EVs as potential therapeutic agents. The writing is clear and pleasing to read.
I was wondering if authors could include any quantitative data in pertaining studies, e.g. EV concentrations in milk, how much cytokine and miRNA in milk-derived EVs, and provide slightly more in depth details in terms of the relationship of milk-derived EVs/milk consumptions and diseases, e.g. Line 140-141.
Responses:
Thank you for your valuable suggestions.
We do not provide specific quantitative data on the concentration of EVs that can be obtained in milk nor the cytokines and miRNAs in EVs.
First, as described in the article, different processing and different EV isolation methods affect the amount of EVs obtained from milk, making it difficult to say how many EVs are actually present in a given volume of milk initially. Nevertheless we now provide the results of Nanosight measurements in the manuscript, revealing that the mode sizes of MEVs in healthy and mastitis groups are 133 nm and 128 nm, respectively. In addition, the concentrations of MEVs were found to be around 6.38×1011 ± 9.56×109 particles/mL in the healthy group and 4.07×1011 ± 1.32×1010 particles/mL in the mastitis group.
The case of cytokines and miRNAs is, however, more complex. Cytokines and miRNAs reflect individual physiological conditions. Therefore, we can only make a general analysis to prove which cytokines and miRNAs are likely to be carried by EVs. Similarly, different processing procedures and detection methods will greatly affect the final measurement results. Therefore, we are unfortunately unable to provide specific and reliable benchmark data. For further relevant information please refer to: Colella, A. P., Prakash, A., & Miklavcic, J. J. (2024). Homogenization and thermal processing reduce the concentration of extracellular vesicles in bovine milk. Food Science & Nutrition, 12, 131–140. https://doi.org/10.1002/fsn3.3749
As for the relationship between EVs or milk consumption and cancerous or inflammatory diseases milk-derived EVs generally are considered to have anti-inflammatory and anti-cancer effects based on current literature. However, more subtle side-effects including the potentially enhanced risk of metastasis formation have also been reported, as mentioned in lines 364 and 365.
Reviewer 2 Report
Comments and Suggestions for Authors
The article “Dairy: Friend or foe? Bovine milk-derived extracellular vesicles and autoimmune diseases” (manuscript ID: ijms-3248724) is a review paper presenting a large amount of literature that associates bovine milk-derived extracellular vehicles (MEVs), which might eventually be administered to humans either as therapeutics per se (presumably due to specific cytokines or microRNAs they contain) or as putative delivery carriers of “exogenous” therapeutics, with autoimmune diseases. As the authors have stated, the field is rather new and there is still much information missing and several issues that should be further elucidated and better understood. The manuscript presents a great deal of relevant literature articles and, to this end, can be accepted for publication, after minor revision.
Comments:
Line 27, “EVs have biological effects on adjacent and remote recipient cells…”: “recipient” might change into “target”
Line 41, Table 1, and line 102, Table 2: A better formatting might help clearly distinguish the data shown in the different rows.
Line 92, Legend of Figure 1B: The length of the scale bar (500.0 nm, not clearly visible in the Figure) might be added here.
Line 97, Figure 1B: This should be properly placed between line 85 and line 86.
Line 155, “2.1. Autoimmune diseases and cytokines” should be corrected to “3.1. Autoimmune diseases and cytokines”
Line 224, “2.2. Autoimmune diseases and miRNAs” should be corrected to “3.2. Autoimmune diseases and miRNAs”
Lines 357-392, “Challenges and future research”: It might be helpful to specially mention how EVs isolated from milk of animals with mastitis may affect/endanger putative therapeutic applications of MEVs.
Lines 362-363, “Then, the composition of MEVs can be influenced by the host’s physiological state…”: Please, define “host” in this context.
Line 390, “consideration of the physiological conditions of the host and the recipient…”: It might be better to change “host” into “donor”
References: Some citations are incomplete, e.g., 15, 41, 42, 49, 59, 60, 84, 105.
Author Response
Comments:
The article “Dairy: Friend or foe? Bovine milk-derived extracellular vesicles and autoimmune diseases” (manuscript ID: ijms-3248724) is a review paper presenting a large amount of literature that associates bovine milk-derived extracellular vehicles (MEVs), which might eventually be administered to humans either as therapeutics per se (presumably due to specific cytokines or microRNAs they contain) or as putative delivery carriers of “exogenous” therapeutics, with autoimmune diseases. As the authors have stated, the field is rather new and there is still much information missing and several issues that should be further elucidated and better understood. The manuscript presents a great deal of relevant literature articles and, to this end, can be accepted for publication, after minor revision.
Line 27, “EVs have biological effects on adjacent and remote recipient cells…”: “recipient” might change into “target”
Line 41, Table 1, and line 102, Table 2: A better formatting might help clearly distinguish the data shown in the different rows.
Line 92, Legend of Figure 1B: The length of the scale bar (500.0 nm, not clearly visible in the Figure) might be added here.
Line 97, Figure 1B: This should be properly placed between line 85 and line 86.
Line 155, “2.1. Autoimmune diseases and cytokines” should be corrected to “3.1. Autoimmune diseases and cytokines”
Line 224, “2.2. Autoimmune diseases and miRNAs” should be corrected to “3.2. Autoimmune diseases and miRNAs”
Lines 357-392, “Challenges and future research”: It might be helpful to specially mention how EVs isolated from milk of animals with mastitis may affect/endanger putative therapeutic applications of MEVs.
Lines 362-363, “Then, the composition of MEVs can be influenced by the host’s physiological state…”: Please, define “host” in this context.
Line 390, “consideration of the physiological conditions of the host and the recipient…”: It might be better to change “host” into “donor”
References: Some citations are incomplete, e.g., 15, 41, 42, 49, 59, 60, 84, 105.
Responses:
Thank you for your detailed and valuable suggestions. We have made modifications and marked them in the article according to your suggestions. We hope to have reached a satisfactory level with these modifications.
Regarding your suggestion: It might be helpful to specially mention how EVs isolated from milk of animals with mastitis may affect/endanger putative therapeutic applications of MEVs. We are sorry that we cannot provide relevant data at present yet as original research is still ongoing in our labs. Current literature mainly focuses on the proteome analysis of mastitis-affected MEVs and the application of EVs to diagnose, treat, and perform disease mechanism analysis in mastitis, but has not studied how EVs from mastitis may affect the human body or endanger putative therapeutic applications. Ji, Z.-H.; Ren, W.-Z.; Wu, H.-Y.; Zhang, J.-B.; Yuan, B. Exosomes in Mastitis—Research Status, Opportunities, and Challenges. Animals 2022, 12, 2881. https://doi.org/10.3390/ani12202881
Our research group is currently producing interim results and hope to publish them soon.